# UV Effect on Human Anterior Lens Capsule Macro-Molecular Composition Studied by Synchrotron-Based FTIR Micro-Spectroscopy

**DOI:** 10.3390/ijms22105249

**Published:** 2021-05-16

**Authors:** Xhevat Lumi, Tanja Dučić, Martin Kreuzer, Marko Hawlina, Sofija Andjelic

**Affiliations:** 1Eye Hospital, University Medical Centre, 1000 Ljubljana, Slovenia; xhlumi@hotmail.com (X.L.); marko.hawlina@gmail.com (M.H.); 2CELLS-ALBA, Carrer de la Llum 2-26, 08290 Cerdanyola del Valles, Barcelona, Spain; tducic@cells.es (T.D.); mkreuzer@cells.es (M.K.)

**Keywords:** UV irradiation, lens capsule, lens epithelial cells, oxidative stress, FTIR, synchrotron light, macro-molecular composition, cataract

## Abstract

Ultraviolet (UV) irradiation is an important risk factor in cataractogenesis. Lens epithelial cells (LECs), which are a highly metabolically active part of the lens, play an important role in UV-induced cataractogenesis. The purpose of this study was to characterize cell compounds such as nucleic acids, proteins, and lipids in human UV C-irradiated anterior lens capsules (LCs) with LECs, as well as to compare them with the control, non-irradiated LCs of patients without cataract, by using synchrotron radiation-based Fourier transform infrared (SR-FTIR) micro-spectroscopy. In order to understand the effect of the UV C on the LC bio-macromolecules in a context of cataractogenesis, we used the SR-FTIR micro-spectroscopy setup installed on the beamline MIRAS at the Spanish synchrotron light source ALBA, where measurements were set to achieve a single-cell resolution with high spectral stability and high photon flux. UV C irradiation of LCs resulted in a significant effect on protein conformation with protein formation of intramolecular parallel β-sheet structure, lower phosphate and carboxyl bands in fatty acids and amino acids, and oxidative stress markers with significant increase of lipid peroxidation and diminishment of the asymmetric CH_3_ band.

## 1. Introduction

Besides the skin, the eye is the organ most exposed to sunlight as well as artificial light sources [1]. Factors such as ultraviolet (UV) irradiation and oxidative stress can induce both in vivo and in vitro cataract formation, which is the leading cause of blindness worldwide [2,3]. The UV light radiation has been shown to be an important factor in cataractogenesis [3]. A common photobiological scheme classifies UV irradiation into three divisions according to their wavelength: UV C (100–280 nm), UV B (280–315 nm), and UV A (315–400 nm). UV B and UV C are the most responsible for photochemical reactions although UV C represents the most biologically damaging range of UV radiation [4,5]. UV C is absorbed in the atmosphere before reaching the earth’s surface but it comes from artificial sources such as germicidal UV lamps or arc welding and therefore constitutes a potential occupational hazard [6]. The artificial light source high-energy UV C irradiation effect is less studied. The damaging effect of UV irradiation in the context of cataractogenesis on lens γ and α crystalline proteins has been reported [4,7,8].

Lens epithelial cells (LECs) in a monolayer under the anterior lens capsule (LC) are the first lens cells exposed to diverse environmental factors such as UV irradiation as well as oxidative stress during metabolic changes [9]. The LEC injury induced by UV irradiation starts with membrane permeability change, which results in losing the lens ions homeostasis [10,11]. It was also shown that UV A could induce disruption of gap junction intercellular communication in LECs [12]. After exposure to UV B radiation, LECs show imbalances in the repair of DNA damage [13]. The consequence may be the notable damage to cell structures such as damage of DNA, RNA, and proteins [10,11]. This UV-induced LECs damage triggers apoptosis on these cells that further leads to cataract formation [14,15]. Human LEC apoptosis is thought to be critical moment in the process of cataract development [2,16]. Although LECs are supplied with machinery to combat cataractogenic trauma, any change in the lens epithelium may proceed further in the remaining part of the lens, consisting of the lens fiber cells, and leading to a cataract [17]. 

In this study, we analyzed the protein conformal changes and changes on lipids and nucleic acids in LECs of both UV C-irradiated and control, non-irradiated LCs obtained from patients without cataract. Bio-macromolecular compounds changes of UV C-irradiated LCs were compared to the non-irradiated LCs. To assess LECs bio-macromolecules involved in UV C-induced changes and to provide their molecular fingerprint, we used synchrotron radiation-based Fourier transform infrared (SR-FTIR) micro-spectroscopy. In addition to spectroscopical analysis, several areas of control LC were investigated by SR-FTIR microscopy in order to locate possible changes within individual anterior lens epithelium. 

SR-FTIR micro-spectroscopy is a vibrational spectroscopic technique that is a potent tool for the cell components analysis, such as nucleic acids [18], proteins [19], and lipids [20]. Spectral data analysis provides qualitative and quantitative information of cell compounds on the basis of peak’s shifts, bandwidths, and band intensities. To the best of our knowledge, this is the first SR-FTIR micro-spectroscopy analysis of the UV C effect on human LCs from patients without cataract. Since UV C rays have the shortest wavelengths from the UV spectrum and are absorbed by the ozone layer before reaching the earth’s surface, this is purely basic science, methodology-oriented research without known clinical implications.

## 2. Results

By using a novel setup at the ALBA synchrotron and the SR-FTIR micro-spectroscopy endstation MIRAS, we evaluated and compared the complete bio-macromolecular information in UV C-irradiated and non-irradiated LCs. Here, we compared the UV C-irradiated non-cataractous LC (5U, C, Ba) with non-irradiated non-cataractous LCs (3, C, Ba and 6, C, Ba). The results are shown in Figure 1 and Figure 2. 

Figure 1 shows protein components, i.e., amide I and amide II regions including the ester groups (1485–1760 cm^−1^). A shift in the band of the β-sheet structure is shown in Figure 1A, which is more visible in the second derivative of the spectra, as shown in the Figure 1D. We observed a shift from 1630 cm^−1^ for one non-irradiated LC, 1633 cm^−1^ for another non-irradiated LC, to 1635cm^−1^ for UV C-irradiated LC. The PC1 showed a strong maximum at 1623 cm^−1^ and a minimum 1668 cm^−1^. The PC2 displayed maxima at 1518, 1652, and 1686 cm^−1^. The main difference was at 1623 cm^−1^, connected to protein cross β- structure [21] and a shift to higher wavenumbers connected to formation of parallel β- structure upon UV C irradiation. 

Figure 2 displays the nucleic acids (A) and lipid areas (D) of the FTIR spectral range. The UV C-irradiated and non-irradiated, non-cataractous LCs differed mainly in the PC1 component, as shown by the scores plot (Figure 2B). The PC1 showed a minimum at 1080 cm^−1^, corresponding to differences in carbohydrates and the symmetric stretching P=O bonds (~1080 cm^−1^) in the phosphodiester group, being more present in UV C-irradiated LC (Figure 2C).

The region between 950 and 1485 cm^−1^ corresponding to the nucleic acids and carbohydrates (Figure 2A) shows that the most pronounced differences were in the asymmetric band at 1240 cm^−1^. The UV C-irradiated sample showed decreased infrared absorbance, which was confirmed by the PCA (Figure 2B,C). The loadings of PC1 and PC2 had strong contributions at 1240 cm^−1^ (maximum), corresponding to asymmetric P=O band in DNA, which was more pronounced in non-irradiated LCs. The band at ~1400 cm^−1^ corresponding to the COO- symmetric stretching mode [22] was more present in non-UV C-irradiated LC. The band at 1110 cm^−1^ corresponded to C–O band in ribose of RNA and was more pronounced in non-irradiated LCs.

The lipid spectral area is presented in Figure 2D–F. The main difference was in the CH_3_ asymmetric vibration, which was drastically decreased in the UV C-irradiated LC. PC1 pointed to differences at 2970 and 2852 cm^−1^, whereas differences were at 2881 cm^−1^ for PC2.

The oxidative stress was estimated by following the lipid peroxidation and using the ratio of lipidic bands: asymmetric vibrations of CH_2_ and CH_3_ (ν_as_ CH_2_/ν_as_ CH_3_ i.e., A2925/A2960), as shown in Figure 3. This ratio was significantly higher in UV C-irradiated LC (Figure 3A). Moreover, the ratio of carbonyl groups to asymmetric bands of CH_2_ and CH_3_ (A1740/A2960 + 2925) also showed the significant differences in the UV C-irradiated LC (Figure 3B).

As one typical example of FTIR imaging, Figure 4 demonstrates the visible images of a LC (sample 3, C, Ba) with chemical FTIR images overlapped in color scales. The colors represent the distribution of the integrated areas of the vas (P=O) band at 1238 cm^−1^ (Figure 4B), the β-sheet contribution at 1625 cm^−1^ (Figure 4C), and the vas (CH_3_) at 2960 cm^−1^ (Figure 4D) along the sample. While the absorbance bands for the β-sheet and vas (CH_3_) bands showed similar distributions along the sample, the vas (P=O) band showed an inverted profile. 

## 3. Discussion 

LECs regulate most of the homeostatic functions of the crystalline lens. They represent a more important part of the mechanisms for energy production, antioxidative mechanisms, and biochemical transport for the whole lens [23,24]. Any change in the human anterior LC macro-molecular composition occurring upon UV exposure is of a high relevance for understanding UV-induced cataractogenesis. Among the several risk factors related to cataract formation, UV radiation is an important and preventable one [1]. 

Not much is known about the LEC compound (proteins, lipids, DNA) changes associated with UV C-induced cataract development. As the lens epithelium on the basal lamina is the first physical and biological barrier in the lens between the aqueous humor and the lens fiber cells and it is metabolically the most active part of the lens, which sustains its physiological condition, understanding the effect of UV irradiation is of great importance. 

In the present study, we evaluated the bio-macromolecule changes, such are protein conformation as well as lipids, nucleic acids, and carbohydrates in UV C-irradiated and control, non-irradiated human LCs without cataract. We then further compared the compound changes in UV C-irradiated LC to the non-irradiated LCs. To the best our knowledge, validation of the cell compounds in human LC upon UV C irradiation on the level of single cells has not been analyzed by SR-FTIR up until now. 

FTIR spectroscopy has already been used for the study the whole crystalline lens [25]. We have also previously used FTIR micro-spectroscopy to study the cell compounds of LECs in human nuclear and cortical cataract types [21]. In a previous study, we have also analyzed the structural organization of the human anterior LECs by the complementary use of scanning electron microscopy, transmission electron microscopy, and confocal microscopy, each of them showing the same morphological features, extensions, and entanglements of the LEC cytoplasmic membrane at the border with the basal lamina, while the basal surface of the LECs increased [26].

The results of the present study show significant effect of UV C irradiation on human anterior LC macro-molecular composition. The UV irradiation had a strong effect on the LEC protein conformation. UV light can induce the radiation damage in primary, secondary, tertiary, and quaternary structure of LEC proteins [27]. In UV C-irradiated LCs of our patient without cataract, we found protein formation of intramolecular parallel β-sheet structure. In contrast, the non-UV-irradiated LCs showed higher contributions for the intermolecular cross β-sheet structure. The β-sheet secondary structure of crystalline lens proteins has been linked to the amyloid formation. Amyloid β-sheets have a range from 1611 to 1630 cm^−1^, whereas native β-sheet proteins produce amide I′ peak clustering between 1630 and 1643 cm^−1^ [28], which has been connected to the UV irradiation as a sign of degradation of the control mechanisms [29]. A shift to higher wavenumbers connected to formation of parallel β-structure was observed in UV C-irradiated LC. Previous studies have shown that crystallin proteins in the crystalline lens have large amounts of native β-sheets [30]. In fact, the majority of water-soluble proteins of the lens are represented by crystallins [31]. Crystallins are structural proteins in mammal crystalline lens comprised of three families: α, β, and γ [32]. Chaperone-like activity of α-crystallin in the protection of other lens proteins such as γ- and β-crystallin against UV irradiation has been demonstrated [32]. The induction to overexpress α-crystallin in the lens has been shown to increase its resistance to photochemical and other stress conditions and also to protect LECs against apoptosis [31]. The formation of intramolecular parallel β-sheet structure can therefore be a protective response of LECs against induced stress conditions.

Oxidative stress is thought to be a common environmental factor in most age-related cataracts [33,34]. Inner cell metabolism together with external or environmental factors could induce oxidative stress, which has been suggested to enhance lipid peroxidation and cellular membrane damage in LECs [14,35,36]. In UV-induced lens damage, one of the early events is the lens epithelial lipid peroxidation [11]. In this study, we found that oxidative stress was significantly higher in UV C-irradiated LC. Oxidative stress markers showed significant increase of lipid peroxidation with diminishment of the asymmetric CH_3_ band after UV C irradiation. In vitro and in vivo studies of cataract formation suggest that the largest effect is through photochemical generation of reactive oxygen species (ROS) and consequent oxidative stress to the lens cells [37]. ROS levels can increase importantly during environmental stress such as UV irradiation. Cataract formation takes place when the rate of ROS production in the lens is greater than the rate of ROS removal [38]. LECs are recognized and studied as the targets of oxidative stress acting on lipids. It was shown that oxidative stress leads to changes in membrane composition in human LECs [39]. Moreover, lipid composition of lens epithelial membranes is different from fiber cell membranes. By Raman spectroscopy, researchers found that LEC membranes contain more phosphatidylcholine and less sphingolipids than cell membranes of fiber cells [40]. Alterations in the defined composition and structure of cell membrane could result in membrane function changes and disruption of the homeostasis of the cell [40]. LECs have high metabolic activity, which makes them subject to oxidative damage. Mitochondrial induced oxidative stress suggests an increase in ROS driven out from UV-exposed respiratory chain, leading to phospholipid hyperperoxidase that, in turn, enhances lipid peroxidation and cellular membrane damage [14,35,36]. The fact that the oxidative stress markers showed significant increase of lipid peroxidation after UV treatment is in agreement with our previous work where the oxidative stress and the lipid peroxidation were found to be more pronounced in LECs of cortical cataracts [21]. 

Beside structural changes in cell membrane composition, oxidative stress can induce damage to other cell structures such as damage of DNA, RNA, and proteins [10,11]. In recent years, the role of epigenetic modifications in pathogenesis of cataracts has been proven [41]. Patterns and levels of DNA methylation and histone modification are the most studied epigenetic modifications in the context of gene transcription [42,43]. UV B has been shown to act on LECs of age-related nuclear cataract through coordinated DNA hypermethylation and histone deacetylation [44]. UV B irradiation in rats produced 6-4 photoproducts or cyclobutane pyrimidine dimer photolesions in LECs. Cyclobutane pyrimidine dimers that were used to visualize DNA adducts were particularly prevalent and were repaired slowly if at all [45]. Our results also showed that UV C-irradiated LC had lower phosphate and carboxyl band in fatty acids and amino acids. In non-irradiated LCs, asymmetric phosphate band in DNA is more present, as well as COO- symmetric stretching mode, and moreover C–O band in ribose of RNA. The fact that the lens epithelium is the lens region having the nucleus and DNA, while lens fiber cells close to crystalline lens nucleus are without cell nucleus, reflects the importance of DNA protection in order to protect the health of the lens. It was shown that UV B modulates DNA synthesis on LECs [46]. After exposure to UV B radiation, LECs showed imbalances in the repair of DNA damage, which can induce changes in the levels of certain proteins, including α-crystallin [13]. The rate of telomere shortening is also modulated by oxidative stress and by changed antioxidative defense capacity [47]. DNA lesions triggered by UV irradiation can result in rapid shutdown of RNA synthesis [48], which might lead to structural alterations of different intracellular proteins. [49]. 

In this study was analyzed the effect of UV C irradiation in cellular macromolecules in LEC of LCs provided from patients without cataract. The induced changes after UV irradiation were compared with macromolecules in LEC of LCs without UV irradiation. We found significant alterations in the structure of proteins, nucleic acids, and lipids as well as in the level of oxidative stress after UV C irradiation. To the best of our knowledge, this is the first study of its kind where the differences between UV C-irradiated and non-irradiated LEC in human LC without cataract were analyzed. However, this study has some limitations. The main limitation is that the doses of UV C irradiation applied to the LCs were high. The dose of irradiation would most likely damage the cornea before even reaching the lens. The limitation is also the fact the UV C radiation from the sun does not reach the earth’s surface, and therefore humans can be exposed to UV C radiation only from an artificial source. Nevertheless, we have shown that UV C irradiation triggers similar cataractogenic changes in macromolecules of LEC as UV rays from the rest of the UV light spectrum. We also present proof-of-concept data in support of FTIR as a tool for the cell macromolecule analysis. We hope this study will stimulate further work in this field, also emphasizing the adequacy of the FTIR methodology.

## 4. Materials and Methods

The LCs were collected from routine uneventful cataract surgery performed at the Eye Hospital, University Medical Centre, Ljubljana, Slovenia. Informed consent was provided before surgery for each patient. Tissue collection and processing were performed according to the Declaration of Helsinki. 

LCs were obtained from patients with primary pathology in posterior segment of the eye. The decision to perform combined phacoemulsification and vitrectomy was made in every case in agreement with the patient in order to reduce the possibility of two surgeries in short period of time and to improve early visual rehabilitation. The 5–5.5 mm big circles of the central anterior LCs were carefully removed by continuous curvilinear capsulorhexis. The capsules were dissected so that the anterior portion of the LC (i.e., basal lamina and associated LECs) were isolated from the fiber cells that form the bulk of the lens. 

After collection, each LC was stored in high-glucose medium (DMEM; Sigma, no. 5671, St. Louis, MO, USA) supplemented with 10% FBS and 1% antibiotics (penicillin–streptomycin; Sigma, no. 4333) and transported to the experimental laboratory of the Eye Hospital. Selected LC was exposed to UV irradiation (UV C lamp Sylvania G 30W, Yokohama, Japan) in laminar hood for 60 min (Iskra PIO, LFV 12). The spectral irradiance of the light source (μW/m^2^/nm) at the position where the tissue was manipulated was measured with a radiometric measurement system consisting of a cosine corrector, a quartz fiber, and a diode-array spectrometer, calibrated with a NIST-traceable source (fiber 200 μm/0.22NA, spectrometer Flame, deuterium-halogen source DH-2000, all Ocean Insight, Dunedin, FL, USA). The major peak wavelength was at 250 nm (UV C, 96.8%) with 0.33444 mW/cm^2^ the second peak wavelength was at 310 nm (UV B, 1.8%) with 0.0061465 mW/cm^2^, and the third peak wavelength was at 360 nm (UV A, 1.4%) with 0.0047835 mW/cm^2^.

LCs were then prepared for FTIR studies: they were first rinsed in 5 mL NaCl for 10 min and then placed by gently stretching and plating adherently on circular 13 mm × 0.5 mm barium fluoride (BaF_2_) slides (Crystan Ltd., Dorset, UK) by using micro-dissecting tweezers (WPI by Dumont, Med.Biologie, Germany). After this, the samples were dried under sterile conditions in the laminar flow at room temperature and stored over silica gel prior the measurements at the ALBA synchrotron.

UV C-irradiated LC was obtained from a 65-year-old male without cataract (5U, C, Ba). Control, non-irradiated LCs were from 2 male patients without cataract: aged 73 years (3, C, Ba) and 70 years (6, C, Ba).

### 4.1. Synchrotron Radiation-Based FTIR Micro-Spectroscopy

To assess the organic compounds profiles, we performed measurements at the infrared micro-spectroscopy beamline MIRAS at the ALBA synchrotron light source (Barcelona, Spain) [50]. Although conventional FTIR spectroscopy is a valuable tool for examining larger cell populations in the tissues, the limited brightness of standard infrared light sources generally precludes high spatial (single-cell) resolution measurements compared to synchrotron radiation-based FTIR (SR-FTIR) micro-spectroscopy [51]. All SR-FTIR micro-spectroscopic absorption spectra were collected in transmission mode using the infrared microscope Hyperion 3000 coupled to a Vertex 70 spectrometer (Bruker, Germany), equipped with a liquid nitrogen cooled mercury cadmium telluride (MCT) detector. Each spectrum was acquired after co-adding 128 scans at spectral resolution of 4 cm^−1^. We used the OPUS 8.2 (Bruker, Germany) software package for data collection. From the LC of the patients 5U, 3, and 6, we measured 128, 122, and 111 individual cells, respectively. The spectral analysis was focused on the wavenumber regions of the fingerprint region (950–1485 cm^−1^), i.e., nucleic acids and carbohydrates, and amide I and II (1485–1760 cm^−1^), i.e., proteins and lipids (2800–3000 cm^−1^). Spectra were baseline-corrected and unit vector-normalized in the regions of interest. Data correction and further analysis was performed by using the ORANGE software package (Bioinformatics Laboratory of the University of Ljubljana [52], Version 3.20.1) with the spectroscopy package [53]. The datasets were compared using a principal component analysis (PCA).

The oxidative stress was estimated following the lipid peroxidation by using the ratio of lipidic bands: asymmetric vibrations of CH_2_ and CH_3_ (νas CH_2_/νas CH_3_) (A2925/A2960), as well as by ratio of carbonyl groups to asymmetric bands of CH_2_ and CH_3_ (A1740/A2925 + 2960). In order to achieve the single cell data acquisition and analysis, we acquired spectra of 10 × 10 μm^2^ areas of the tissue by using the aperture of the microscope. 

### 4.2. FTIR Micro-Spectroscopy Imaging

Matrices with 5 × 40 spectra were set with an aperture size of 10 × 10 µm^2^ and a step size of 60 µm (Figure 4). The chemical FTIR images were generated by integrating the spectral regions of interest from 1245 to 1230 cm^−1^ for P=O, from 1630 to 1620 cm^−1^ for v(β-sheet), and from 2965 to 2955 cm^−1^ for vas (CH_3_). FTIR imaging was performed on a non-cataractous LC (3, C, Ba). All spectra were baseline-corrected for the three regions (fingerprint, proteins, and lipids) before integration using OPUS 8.2 (Bruker, Germany). 

## 5. Conclusions

UV irradiation is an important factor in the process of cataract development. The first lens part to be exposed to UV is the anterior LC with LECs. The results of the presented SR-FTIR micro-spectroscopy analysis of the LC compounds after UV C irradiation, even though the sample of UV-irradiated human LCs was small, clearly showed evident differences in comparison to the control, non-irradiated LCs compounds. The UV C irradiation had a strong effect on the LEC protein conformation. In UV C-irradiated LC, we found protein formation of intramolecular parallel β-sheet structure, unlike the nonirradiated LCs that showed higher contribution for the intermolecular cross β-sheet structure. Our results also showed that UV C-irradiated LC had lower phosphate and carboxyl bands in fatty acids and amino acids. In non-irradiated LCs, asymmetric phosphate band in DNA was more present, but also COO- symmetric stretching mode, as well as C–O band in ribose of RNA. We also found that oxidative stress was significantly higher in UV C-irradiated LC. Oxidative stress markers showed significant increase of lipid peroxidation with diminishment of the asymmetric CH_3_ band after UV irradiation. 

Our results obtained by SR-FTIR micro-spectroscopy increased the pale knowledge about the total proteins, lipids, and nucleic acid changes in single cells in lens epithelium related to the UV C light. We have also shown the utility of SR-FTIR micro-spectroscopy for investigating the UV effect on human lenses. We hope this study will help in understanding the UV C-induced cataractogenesis and the role of LECs in preventing it.

## Figures and Tables

**Figure 1 ijms-22-05249-f001:**
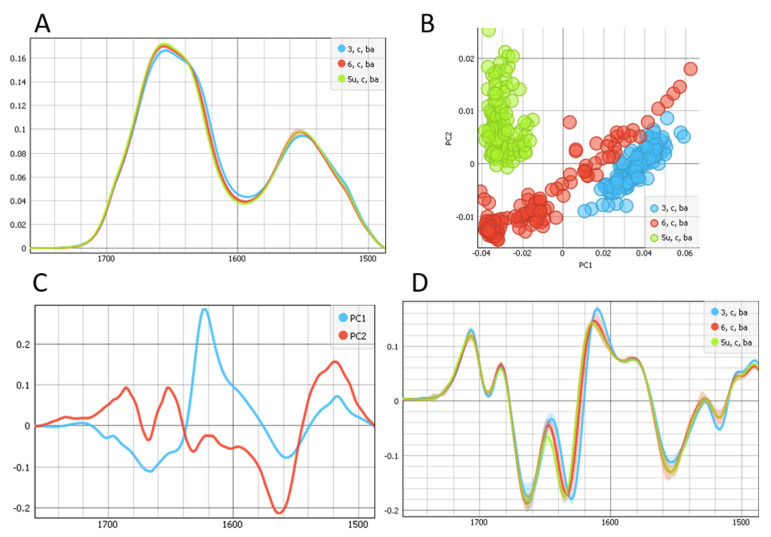
(**A**) Average spectra of the protein and ester regions (1485 until 1760 cm^−1^) of UV-irradiated (green) non-cataractous LC and the two non-irradiated (red and blue) non-cataractous LCs. (**B**) The PCA scores plot denotes the variability associated with the first two components. (**C**) First and second PCA components, PC1 and PC2 score plot. (**D**) The second derivative of average spectra of the protein and ester regions.

**Figure 2 ijms-22-05249-f002:**
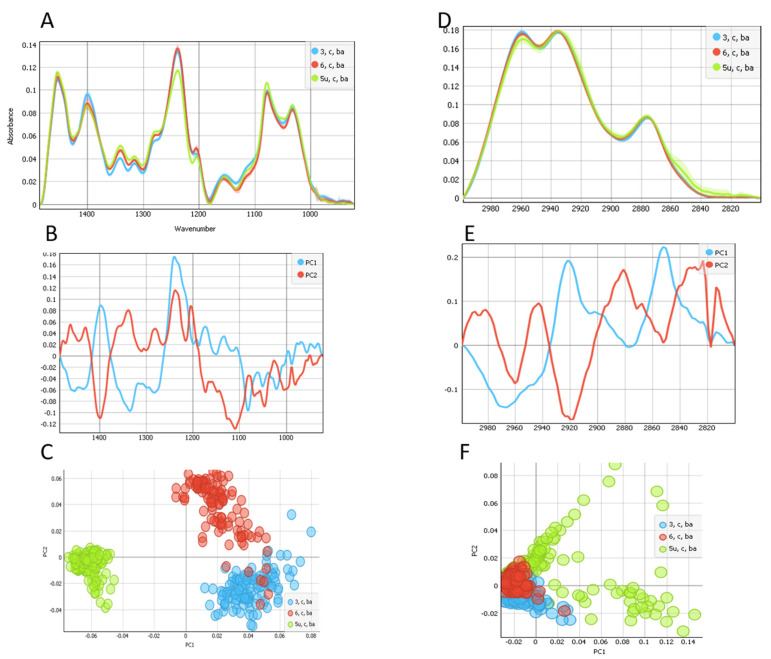
Analysis of the spectral region of the nucleic acids and carbohydrates (**A**–**C**) and lipids (**D**–**F**) of UV-irradiated (green) non-cataractous LC and the two non-irradiated (red and blue) non-cataractous LCs. (**A**) The FTIR average spectra of fingerprint area (950–1485 cm^−1^). (**B**) The PCA score plot denotes the variability associated with the first two components. (**C**) First and second PCA components, PC1 and PC2 score plot. (**D**) The FTIR average spectra of lipid area (2800–3000 cm^−1^). (**E**) The PCA score plot denotes the variability associated with the first two components. (**F**) First and second PCA components, PC1 and PC2 score plot.

**Figure 3 ijms-22-05249-f003:**
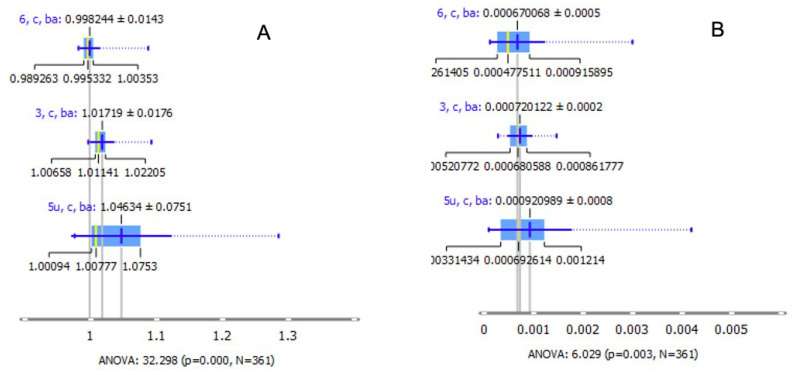
Analysis of oxidative stress markers. Distribution of the ratios between the asymmetric CH_2_ and CH_3_ bands (**A**) ((**A**) for non-cataractous LCs 3, 6, and 5U) and the ratio between the C=O band and the sum of asymmetric CH_3_ and CH_2_ bands ((**B**) for non-cataractous LCs 3, 6, and 5U). Values are presented with the probability density of the data at different values and mean ± SD.

**Figure 4 ijms-22-05249-f004:**
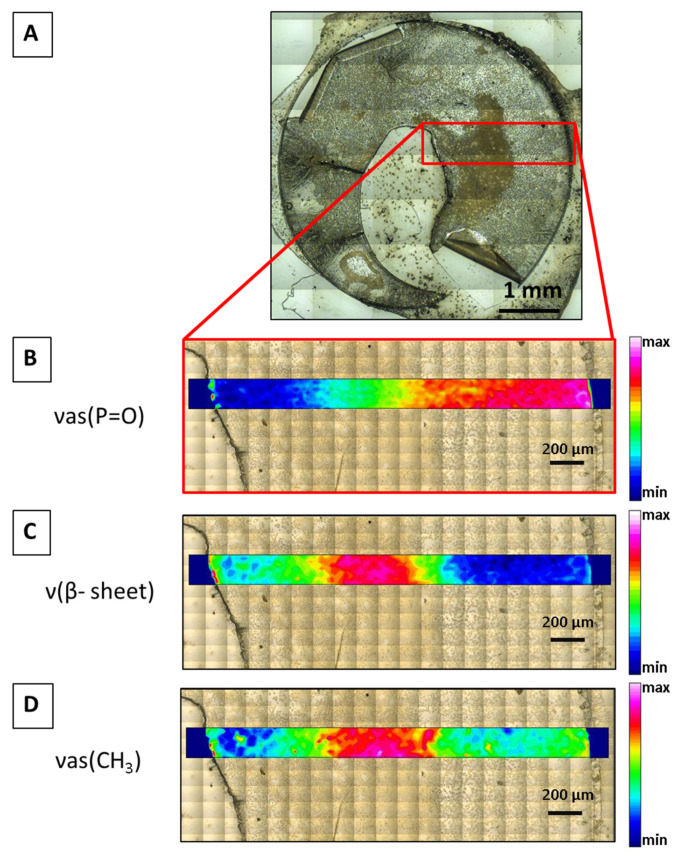
(**A**) Visible image of a non-cataractous LC 3, C, and Ba taken in reflection geometry. Highlighted is the FTIR imaging area. Visible images with higher magnification achieved in transmission geometry are shown below with overlapped chemical FTIR micro-spectroscopy images showing the integrated intensities of the vas (P=O) band at 1238 cm^−1^ (**B**), the β-sheet contribution at 1625 cm^−1^ (**C**), and the *v*_as_ (CH_3_) at 2960 cm^−1^ (**D**).

## Data Availability

The data presented in this study are available in request from the corresponding author.

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
