# Peer review of "UV Effect on Human Anterior Lens Capsule Macro-Molecular Composition Studied by Synchrotron-Based FTIR Micro-Spectroscopy"

_ijms, 2021, doi:10.3390/ijms22105249_

Round 1
Reviewer 1 Report
This study used synchrotron radiation-based Fourier transform infrared spectroscopy to investigate effects of in vitro UV radiation on proteins and lipids present in isolated human lens capsules and epithelia. It is assumed that each sample came from a single lens of each of 5 patients. The samples included a UV irradiated diabetic cataract (82 year old female), a non-irradiated diabetic cataract (61 year old female), a UV irradiated normal lens (65 year old male), and two non-irradiated normal lenses (70 and 73 year old males).
The rationale for studying UV radiation of a diabetic cataract is not clear. Diabetic cataract is not even mentioned in the abstract. It is not clear why normal lenses were removed from three patients, although it is stated that patient consent was obtained. The normal lens is believed to provide a number of protective functions for other eye tissues.
A major weakness of the paper is that no information at all was provided about the method of UV radiation. This would include the range of wavelengths in the UV beam, the peak wavelength, mW/cm2, time of irradiation, J/cm2, and whether any UVC radiation in the beam was filtered out. The human lens does not interact with any solar radiation wavelengths lower than 297 nm. The maximum UVB dose experienced by a human lens outside on a sunny day would be about 0.001 mW/cm2 or about 0.004 J/cm2/hr. If this work employed light with a wavelength less than 297 nm or very high UVB doses, the results would have little physiological relevance.
It’s not surprising that isolated human lens capsule/epithelia exposed to UV light would show major effects on proteins and lipids. However, the relevance to UVB light as a cause of human cortical cataract is not clear.
Reviewer 2 Report
UV irradiation is important factor in the process of cataract development and the first lens part to be exposed to UV is the anterior LCs with LECs. The results from FTIR analysis of the LCs compounds after UV irradiation, even though the sample of UV irradiated human LCs is small, show clearly differences in comparison to the control, non-irradiated LCs compounds. Authors found that UV irradiation induced similar changes in both non-cataractous LC and in diabetic LC with cataract. However, the results in figures 1 and 2 mixed the UV irradiated diabetic and control. In vivo, the diabetic cataract lens is very different from control lens. The result between diabetic and control after UV irradiation should be different unless the lens from patient were at senior ages. If the authors want to show the differences before and after UV irradiation, there is unnecessary to show the sample of diabetic cataract lens.
- I suggest a statement from authors explaining why they did not compare the diabetic cataract lens and control before/after the UV irradiation. Please show the comparison if there were differences between diabetic and control lens before/after the UV irradiation.
- Minor suggestions, the English have some typos. For example, line 291 “we” analyzed.
- Please check the word format/size in the manuscript.
- Please have uniform format in the references.
Round 2
Reviewer 1 Report
This work really has no relevance at all to understanding how UV light may contribute to human cataract formation. If a human was exposed to the UVC dose used in this study, 0.3 mW/cm2 of 250 nm light for 1 hour, cataract would be the least of his worries. That dose of irradiation would most likely severely damage the cornea before even reaching the lens. However, since IJMS is not an eye journal and the other reviewer appeared to be in favor of publishing the study, I will check off "accept" on my review form.
